# Absorptive Desulfurization of Model Biogas Stream Using Choline Chloride-Based Deep Eutectic Solvents

**Edyta Słupek** and **Patrycja Makoś** *

Department of Process Engineering and Chemical Technology, Faculty of Chemistry,
Gdansk University of Technology, G. Narutowicza St. 11/12, 80–233 Gdańsk, Poland; edyta.slupek@pg.edu.pl
* Correspondence: patrycja.makos@pg.edu.pl; Tel.: +48-508997100

**Abstract:** The paper presents a synthesis of deep eutectic solvents (DESs) based on choline chloride (ChCl) as hydrogen bond acceptor and phenol (Ph), glycol ethylene (EG), and levulinic acid (Lev) as hydrogen bond donors in 1:2 molar ratio. DESs were successfully used as absorption solvents for removal of dimethyl disulfide (DMDS) from model biogas steam. Several parameters affecting the absorption capacity and absorption rate have been optimized including kinds of DES, temperature, the volume of absorbent, model biogas flow rate, and initial concentration of DMDS. Furthermore, reusability and regeneration of DESs by means of adsorption and nitrogen barbotage followed by the mechanism of absorptive desulfurization by means of density functional theory (DFT) as well as FT-IR analysis were investigated. Experimental results indicate that the most promising DES for biogas purification is ChCl:Ph, due to high absorption capacity, relatively long absorption rate, and easy regeneration. The research on the absorption mechanism revealed that van der Waal interaction is the main driving force for DMDS removal from model biogas.

**Keywords:** deep eutectic solvents; absorption; biogas; dimethyl disulfide; green solvents; desulfurization

## 1. Introduction

Currently, more and more attention is paid to the production of alternative high-quality fuels, including bio-methane, bio-hydrogen, bio-ethanol, bio-butanol, etc. Biogas is a modern form of bioenergy, which can be obtained in the process of dark fermentation using waste products from various industries and agriculture (agri-food and animal waste) [1–3]. Biogas usually contains 30%–60% *v/v* $CH_4$, 15%–30% *v/v* $CO_2$, 5%–20% *v/v* $N_2$, 1%–10% *v/v* $O_2$, and about 1%–2% *v/v* of other contaminants including $H_2S$, $NH_3$ and numerous of organic compounds [4–6]. Among the organic pollutants, volatile organosulfur compounds (VSCs) are one of the most important biogas impurities group. VSCs include sulfides, disulfides, thiophenes, and thiols, and the concentrations of them are strictly dependent on the raw material used for biogas production [7]. A typical range of VSCs concentration in biogas is shown in Table 1.

**Table 1.** Range concentrations volatile organosulfur compounds (VSCs) in the biogas stream [7].

| VSCs | Concentration (mg/Nm$^3$ biogas) |
|---|---|
| dimethyl sulfide | 1.25–2.76 |
| carbon disulfide | 1.62–5.91 |
| 2-propanethiol | 0.50–1.19 |
| 1-propanethiol | 2.87–19.51 |
| 2-butanethiol | 0.79–4.65 |
| thiophene | 0.16–1.16 |
| methyl propyl sulfide | 0.35–1.76 |
| dimethyl disulfide | 0.39–1.14 |
| dipropyl disulfide | 0.76–3.16 |

Volatile organosulfur compounds are commonly found in many products and waste streams, causing technological and environmental problems [8,9]. VSCs are chemical compounds which are characterized by high toxicity, malodorousness, and high reactivity [10]. During biogas combustion, VSCs converted into toxic sulfur oxides (SOx) react with oxygen and water, forming sulfuric acid ($H_2SO_4$) which corrode the surface in the combustion chamber. Therefore, biogas pre-treatment is necessary to protect engines that convert biogas into energy and in order to protect an environment [11,12].

Currently, there are many technologies to remove impurities from biogas, which are classified into three main categories: physical (e.g., absorption, adsorption, condensation), chemical (e.g., catalytic oxidation, thermal oxidation, ozonation), and biological (e.g., bio-scrubbers, bio-filters, activated sludge) [13–16]. Most of these methods are expensive and complicated, therefore, more and more attention is paid to optimize them in terms of low energy consumption and high efficiency [17–20]. Among the technologies, physical absorption is one of the most attractive methods due to its simplicity, economical, safe, and its potential high impurities removal efficiency from gas steams [21]. The choice of a suitable absorbent is the key to successful impurities removal. "Perfect" absorbent should be characterized by non-toxic character, high absorption capacity, high-boiling point, low vapor pressure, high diffusion coefficient, low prices, and safety [22]. Several solvents that may be useful for the removal of impurities include water, mineral oils, alcohols, ketones, and amines [23–25]. However, in accordance with the trends of "green chemistry", conventional solvents should be replaced by new generation absorbents. Until recently, studies have focused on ionic liquids (ILs) as absorbents due to their unique properties [26]. However, the problems with their toxicity, stability, biodegradability, and expensive synthesis make them less than ideal solvents. Alternatives to ILs are deep eutectic solvents (DESs), which have similar physico-chemical properties. DESs are a mixture consisting mainly of two or three non-toxic and biodegradable compounds that are capable of forming eutectic liquids, based on the specific interaction between hydrogen bond donor (HBD) and hydrogen bond acceptor (HBA) [27,28]. Until now, DESs have been widely used for desulfurization of fuels [9], biomass pre-treatment [29], water and air purification [30,31], in catalysis [32], and analytical chemistry [27,33,34]. Till now, no studies have been in the literature published regarding the use of DESs as absorbents to remove volatile organosulfur compounds from the biogas stream. The studies presented in the literature are limited to the removal of inorganic compounds, i.e., $H_2O$, $CO_2$, $H_2S$, and $NH_3$ from gaseous phases [35–37]. Only a few works refer to their use to remove volatile organic compounds from the gas phase [38,39].

The paper describes the synthesis of DES composed of choline chloride (HBA) and phenol, ethylene glycol, levulinic acid (HBD) in a 1:2 molar ratio. The absorptive desulfurization was optimized on selection terms of DES type, temperature, absorbent volume, initial concentration of dimethyl disulfide (DMDS), and model biogas flow rate. Two most popular desorption methods including adsorption and nitrogen barbotage were investigated. For the better understand the mechanism of DMDS removal, FT-IR and density functional analysis were employed.

## 2. Materials and Methods

### 2.1. Reagends

The research was used reagents choline chloride (ChCl) (purity ≥99%), diethylene glycol (EG) (purity ≥95%), levulinic acid (Lev) (purity ≥98%), phenol (purity ≥99%), dimethyl disulfide (purity ≥98%), silica gel (SG) (grain diameter dp = 40 µm), active carbon (AC) (grain diameter dp = 0.3–0.5 mm), and aluminum oxide (III) (AO) (grain diameter dp = 40 µm) were purchased from Sigma-Aldrich (USA). Compressed gases such as nitrogen (purity N 5,5), air (purity N 5.0) generated by a DK50 compressor with a membrane dryer (Ekkom, Poland), and hydrogen (purity N 5.5) generated by a 9400 Hydrogen Generator (Packard, USA) were used for the preparation of model biogas, regeneration process, and chromatographic analysis.

### 2.2. Apparatures

Gas chromatograph Autosystem XL equipped with flame ionization detector (GC-FID) (PerkinElmer, USA), HP-5 (30 m × 0.25 mm × 0.25 µm) capillary column (Agilent Technologies, USA), TurboChrom 6.1 software (PerkinElmer, USA), and FT-IR Bruker Tensor 27 spectrometer (Bruker, USA) with an ATR accessory and OPUS software (Bruker, USA) were used.

### 2.3. Procedures

#### 2.3.1. Preparation of DES

DESs were synthesized by mixing ChCl (HBA) with Ph, EG, and Lev (HBD) in a 1:2 molar ratio. The mixtures were stirred magnetically at 65 °C until homogeneous liquids were obtained. The liquids were then left to cool spontaneously to room temperature. The physico-chemical properties of the synthesized DES were presented in Table 2.

**Table 2.** Selected physico-chemical properties of deep eutectic solvents (DESs) reported in the literature.

| HBA | HBD | HBA:HBD Molar Ratio | Melting Points (°C) | Density (g/cm$^1$) (25 °C) | Viscosity (cP) (25 °C) | Ref. |
|---|---|---|---|---|---|---|
| | Ph | | −68.9 | 1.10 | 14 | [40] |
| ChCl | Lev | 1:2 | Liquid at RT * | 1.12 | 32 | [41] |
| | EG | | −66.0 | 1.12 | 37 | [42] |

* RT—room temperature.

#### 2.3.2. Absorption Process

The absorption process was prepared by means of the barbotage phenomena. Nitrogen (model biogas stream) was passed through a 20-mL vial containing 5 mL of DMDS. The created mixture (nitrogen–DMDS) was diluted with a nitrogen stream to 1.0 mg/Nm$^3$ (DMDS) concentration. The model biogas stream containing DMDS was passed through an absorption column (total volume 60 mL) containing 50 mL of DES. The total flow of gaseous DMDS and nitrogen was kept constant at 50 mL/min. The concentration of DMDS was monitored at the inlet and outlet of the barbotage column using GC-FID. The processes were carried out for 1200 min. The absorptivity (A) of DMDS in the DES was calculated using Equation (1):

$$A = \frac{C_{in} - C_{out}}{C_{in}} \tag{1}$$

where $C_{in}$—initial DMDS concentration (ppm *v/v*),

　　　$C_{out}$—DMDS concentration after absorption process (ppm *v/v*).

### 2.3.3. Regeneration of DESs

After the absorption process, DESs were regenerated using two popular methods, including nitrogen barbotage and adsorption. Nitrogen barbotage was carried on as follows: 4 mL of DES was barbotaged using nitrogen flow 50 mL/min for 2.5 or 5 h. Three types of adsorbents, i.e., AC, SG, and AO were used in the second type of regeneration process. All adsorbents were activated in a laboratory dryer at 120 °C for 2 h. The 4 mL of DES containing DMDS was mixed with 160 mg and 420 mg adsorbents in a vial. The vials were maintained in a laboratory shaker at 25 °C for 30 min, subsequently centrifuged for 5 min at 7000 rpm, and filtered through a 0.45 μm cellulose filter. The concentration of DMDS in DES (before and after regeneration) was controlled using static headspace coupled to gas chromatography (SHS-GC).

### 2.3.4. Chromatographic Analysis

GC temperature program was 120 °C; injection port temperature was 300 °C carrier gas–nitrogen (2 mL/min); injection mode: split 20:1; detector temperature was 300 °C; detector gases flow rates were hydrogen 40 mL/min and air 400 mL/min. DESs (2 mL) after regeneration were thermostated at 80 °C for 50 min. Then, 0.1 mL of gas phase was introduced into the GC injector. In order to monitor DMDS concentrations during the absorption process, 0.5 mL of model biogas was analyzed.

### 2.3.5. FT-IR Analysis

The following working parameters of FT-IR analysis were used: spectral range: 4000–550 $cm^{-1}$; number of background scans: 256; number of sample scans: 256; resolution: 4 $cm^{-1}$; slit width: 0.5 cm.

### 2.3.6. Theoretical Studies

The molecular structures and interactions between DES and DMDS were optimized using the B3LYP/6-311++G** level of theory with the dispersion corrected computational model using Orca 4.1.1 software package. All configurations were optimized for local minima using frequency calculations. The interaction energy for the gas phase between the DES and DMDS was calculated according to the following expression (Equation (2)):

$$\Delta E = E_{DES-HMDS} - (E_{DES} - E_{HMDS}) \tag{2}$$

where $E_{DES-HMDS}$—total energy of complex DES-DMDS (kcal/mol);

$E_{DES}$—individual energy of DES (kcal/mol);

$E_{HMDS}$—individual energy of HMDS (kcal/mol).

The counterpoise method has also been implemented to estimate the effects of the basis set superposition error (BSSE) on the interaction energy, based on previous studies [43]. Electrostatic potential analysis (ESP) and reduced density gradient analysis (RGD) were performed to the visual interpret the interaction nature in the DES-DMDS complex. Both RDG and ESP analysis were performed using Multiwfn software [44–46]. In order to the graphical presentation of the results, the Visual Molecular Dynamics 1.9.3, the software was used.

## 3. Results and Discussion

### *3.1. Optimization of Absorption Conditions*

Optimization of absorption conditions using deep eutectic solvents was carried out for DMDS as the main representative of volatile organosulfur compounds commonly found in real biogas streams [7,47–50]. The process was optimized in terms of DES type, temperature, the volume of DES, model biogas flow, and initial concentration of DMDS.

### 3.1.1. Kind of DES

The selection of the absorption solvent is particularly important in the absorptive desulfurization process. Three types of DES were tested, including ChCl:Ph, ChCl:Lev, and ChCl:EG in a 1:2 molar ratio (Figure 1). In the experiment, the following pre-selected absorption conditions were used: 50 mL of DES; initial concentration of DMDS 1 mg/Nm$^3$, 25 °C temperature; model biogas flow rate 50 mL/min. Among the investigated DESs, ChCl:Ph shows the best absorption efficiency. After 800 min, the absorptivity value for ChCl:Ph was below 0.3 and then increased rapidly over the next 300 min, which indicates DES saturation. The saturation time of the other two DES was 600 and 800 min for ChCl:EG and ChCl:Lev, respectively. It can be noticed that the best result was obtained for DES which has the lowest viscosity value. As the viscosity increased, both the absorption capacity and absorption rate decreased. In DES with higher viscosity, the mass transfer is hindered, therefore it is preferable to use solvents with the lowest viscosity.

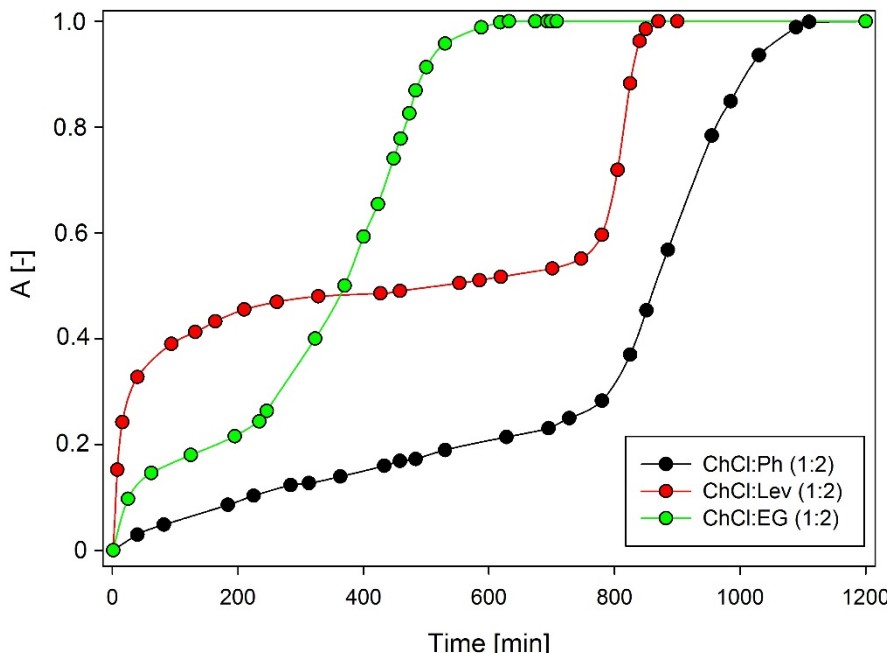

**Figure 1.** Dimethyl disulfide (DMDS) absorption curves using DES under optimum conditions.

It is worth noting that the viscosity of the DES increases when the amounts of hydroxyl groups in HBD increases. The existence of extra hydroxyl groups creates a more extensive hydrogen bond network which results in lower mobility of free species within the DES [28]. However, many other factors (except viscosity) also influence the DMDS removal efficiency, therefore in Section 3.2., the mechanism of absorptive desulfurization of model biogas was explained.

### 3.1.2. Volume of DES

In the studies, three-volume of DES in the range of 15–50 mL/min were investigated (Figure 2a). In the studies, other operating parameters, i.e., inlet concentration DMDS 1 mg/Nm$^3$, nitrogen flow rate 50 mL/min, and the temperature of process 25 °C, were constant. The results showed that with the increase in DES volume, the absorption efficiency increased significantly from 695 min to 1200 min. This is due to the fact that as the volume of the absorbent increases, the contact time between the gas phase and the liquid increases, due to which the saturation time is longer. Similar results were obtained in the work [51,52].

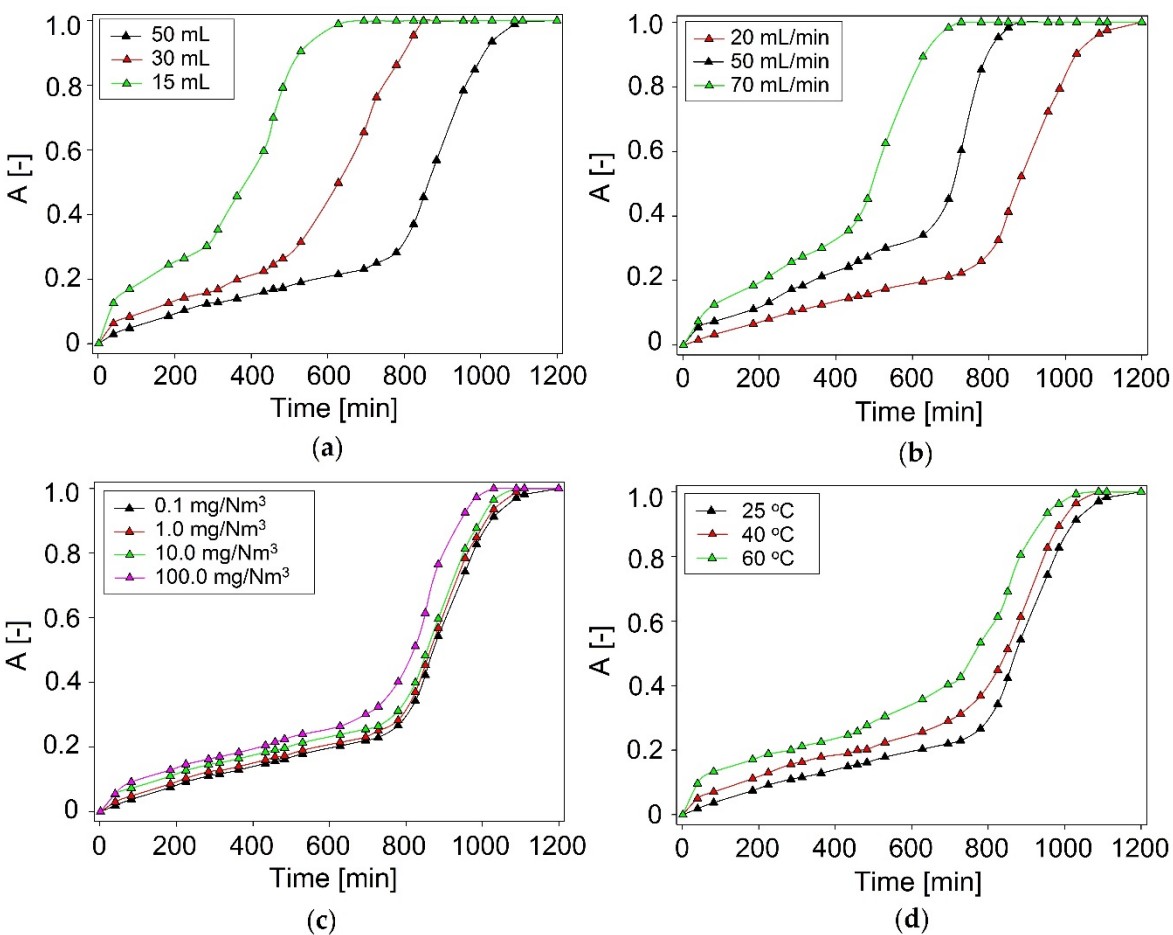

**Figure 2.** Experimental breakthrough curves of DMDS absorption with ChCl:Ph at different: (**a**) volume of DES; (**b**) biogas flow rate; (**c**) initial concentration of DMDS; (**d**) temperature.

### 3.1.3. Model Biogas Flow Rate

The next studied parameter was model biogas flow rate in the range of 20–70 mL/min (Figure 2b). From an industrial point of view, the flow rate of biogas should be as high as possible. The results indicate that the flow rate has a large impact on the overall DMDS capture process. As the flow rate increases from 20 to 70 mL/min, the DES saturation time is reduced from 1200 to 729 min. Similar results were obtained in the research [51–53]. This is due to the fact that as the flow velocity increases, the contact time of the polluted gas stream with the absorbent is reduced, which adversely affects the DMDS absorption process. However, only a slight change in saturation time is observed between the flow of 20 and 50 mL/min. Therefore, a 50 mL/min model biogas flow rate was considered as optimum value.

### 3.1.4. Initial Concentration of DMDS

In the studies, the initial concentration of DMDS in the range of 0.1–100 mg/Nm$^3$ was investigated (Figure 2c). The results indicate that the DMDS absorption efficiency remains fairly stable despite an increase in DMDS inlet concentration. This valuable result shows the ability of ChCl:Ph to the removal of DMDS in varying concentrations from real biogas steam which makes it desirable from an industrial point of view. Similar results were obtained in the work [54].

### 3.1.5. Temperature

Three temperature values, i.e., 25, 40, and 60 °C, were chosen to assess the influence of temperature on DMDS absorption behavior (Figure 2d). Theoretically, an increase in temperature affect the decreases

in DES viscosity and hence gas transfer rate are improved. The absorption curve reveals that by increasing DES temperature, the solubility of DMDS decreases. The decrease in solubility can be explained by the fact that the gas absorption process is normally exothermic. Therefore, the preferred temperature using DES is 25 °C, because in using this temperature the longest effective purification time for the biogas stream is achieved, lasting up to 1200 min. When the temperature is increased to 60 °C, the absorption time is reduced to 1010 min. Consequently, the absorption process can be performed at room temperature with minimal energy consumption. Similar results were obtained at work [55].

### 3.2. Mechanism of Absorption

### 3.2.1. FT-IR analysis

The experimental research on the mechanism of the absorption process was performed by FT-IR analysis. The spectra of pure DESs were compared with pure DMDS, and DES-DMDS complex spectra (Figure 3a–c). All characteristic bands that can be attributed to DMDS (2909.32 cm$^{-1}$—δs (CH) stretch, 1411.73 cm$^{-1}$—δas (CH$_3$) def., 1302.68 cm$^{-1}$—δs (CH$_3$) def., 692.82 cm$^{-1}$—C-S stretch, 540.96 cm$^{-1}$—S-S stretch) are visible in the ChCl:Ph-DMDS spectrum [56], which indicates the creation of the DES-DMDS complex (Figure 3a). Theoretically, in the absorptive DMDS removal process, both sulfur atoms can act as a donor in S–H···π and as an acceptor in O–H···S and C–H···S interactions [57–59]. However, on the FT-IR spectra, there are no shifts corresponding to this type of interaction. Therefore, other interactions must play a key role in the DMDS absorption process. Similar results were also obtained for ChCl:Lev (Figure 3b) and ChCl:EG (Figure 3c).

### 3.2.2. Molecular Modeling

In order to better understand the mechanisms of DMDS removal from the gas phase using DES, the density functional theory (DFT) was applied. For this purpose, the most probable and stable configurations in the gas phase of ChCl:Ph-DMDS, ChCl:Lev-DMDS, and ChCl:EG-DMDS was geometry optimized at the B3LYP/6-311++G** level of theory (Figure 4). The results indicate that in all complex, nonbonded interaction exists between choline and chloride atom (O-H···Cl), which can be identified as strong hydrogen bond because of short distance (below 2.5 Å) [60]. Hydrogen bonds also occur between the Cl atom and two phenol molecules such as Cl···H-O (2.32 Å) and Cl···H-O (2.44 Å) in the ChCl:Ph-DMDS complex, between Cl atom and carboxylic group of levulinic acid (Cl···H-OOC 2.07 Å), carboxylic groups of two levulinic acid molecules (OH···OH 1.88Å) in ChCl:Lev-DMDS, and between the Cl atom and hydroxyl group of one EG molecule (Cl···H-O 2.13 Å) as well as between both EG molecules (OH···HO 2.14 Å) in ChCl:EG-DMDS complex. However, between DMDS and all DES, there are no hydrogen bonds and only weak electrostatic bonds occur.

Electrostatic potential analysis (ESP) was used for visualization of total charge distribution and relative polarity of the studied DMDS structure and DESs-DMDS complexes. The ESP of DMDS, ChCl:Ph-DMDS, ChCl:Lev-DMDS, and ChCl:EG-DMDS are mapped onto their electron densities in Figure 5. The results indicate that the electropositive areas are around the hydrogen atoms whereas the electronegative area is around sulfur atoms, in the DMDS structure. In the ChCl:Ph-DMDS complex, a large electropositive area is around the nitrogen atom and electronegative areas are around Cl, O, and S atoms. During the absorption process, the electronegative region located around sulfur atoms in the DMDS molecule interacts with the electropositive area located around the nitrogen atom in the ChCl:Ph molecule. These interactions provide efficient DMDS removal from the model biogas. On the other hand, it can be concluded that HBA in the DES molecule has the greatest impact on the purification process. This phenomenon will be investigated in subsequent works. Similar results can be found in ChCl:Lev-DMDS and ChCl:EG-DMDS complex.

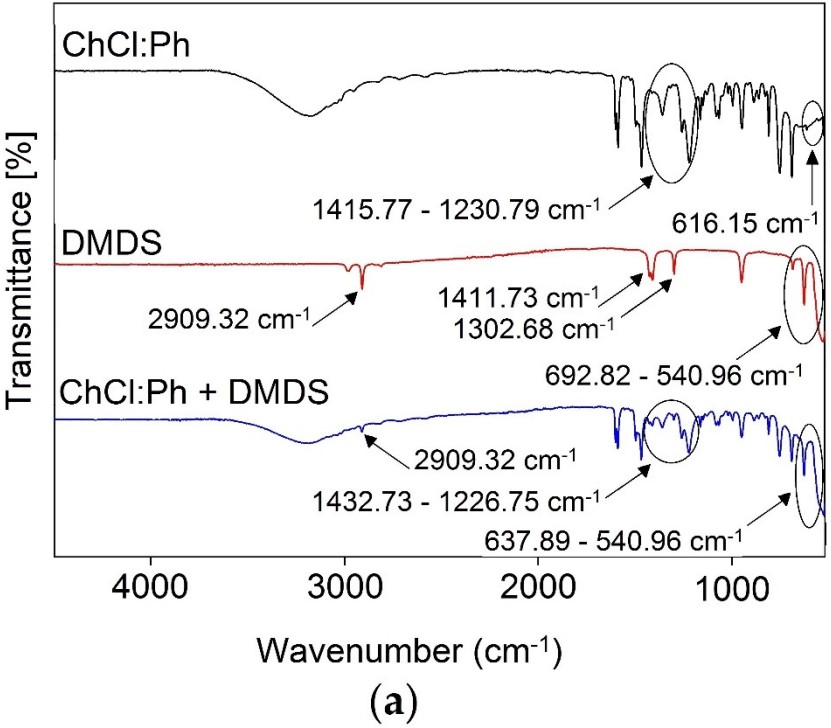

(**a**)

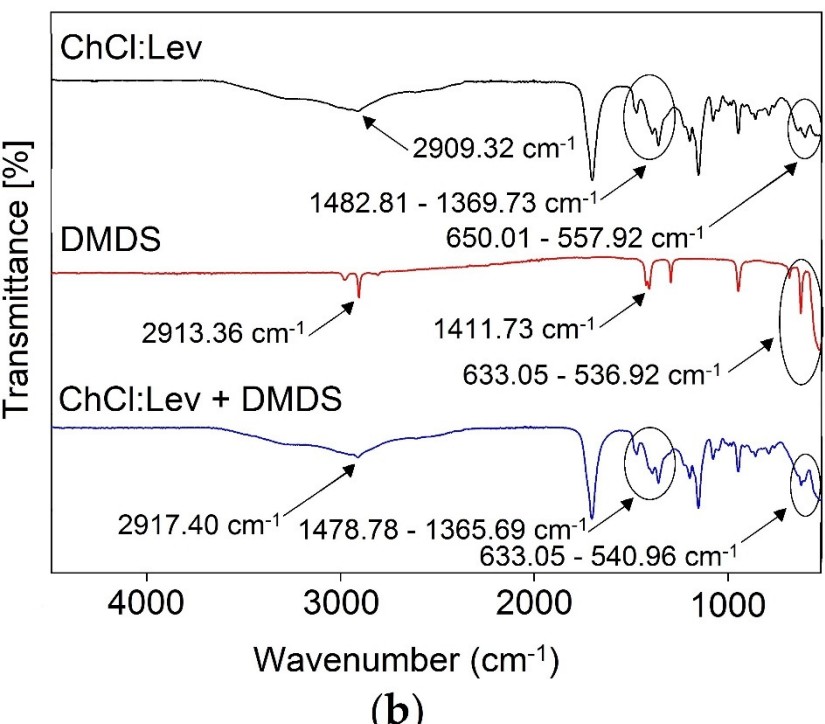

(**b**)

**Figure 3.** *Cont.*

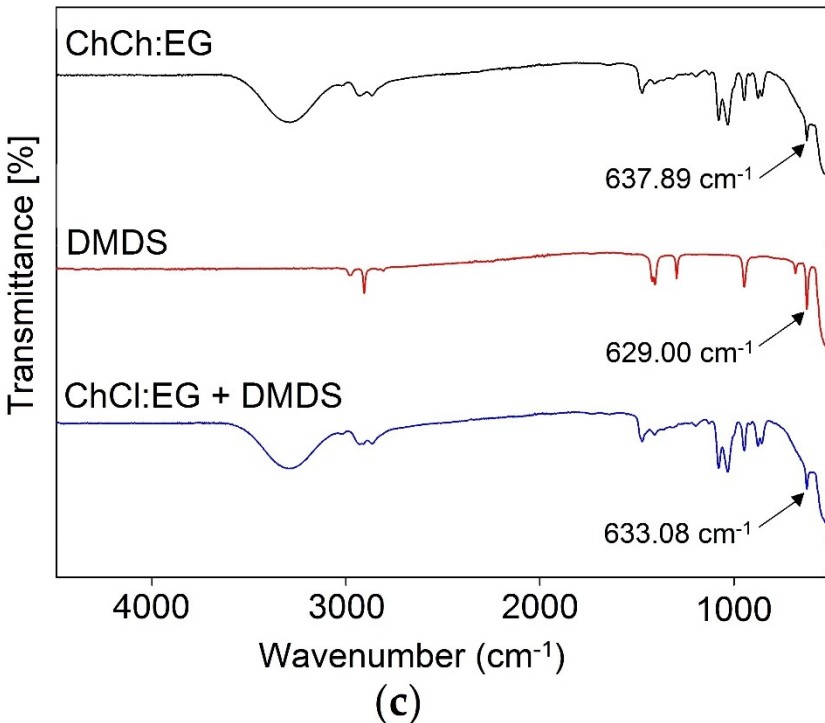

**Figure 3.** FT-IR spectra of: (**a**) pure ChCl:Ph, pure DMDS, and DES after the DMDS absorption process; (**b**) ChCl:Lev, pure DMDS, and DES after the DMDS absorption process; (**c**) pure ChCl:EG, pure DMDS, and DES after the DMDS absorption process.

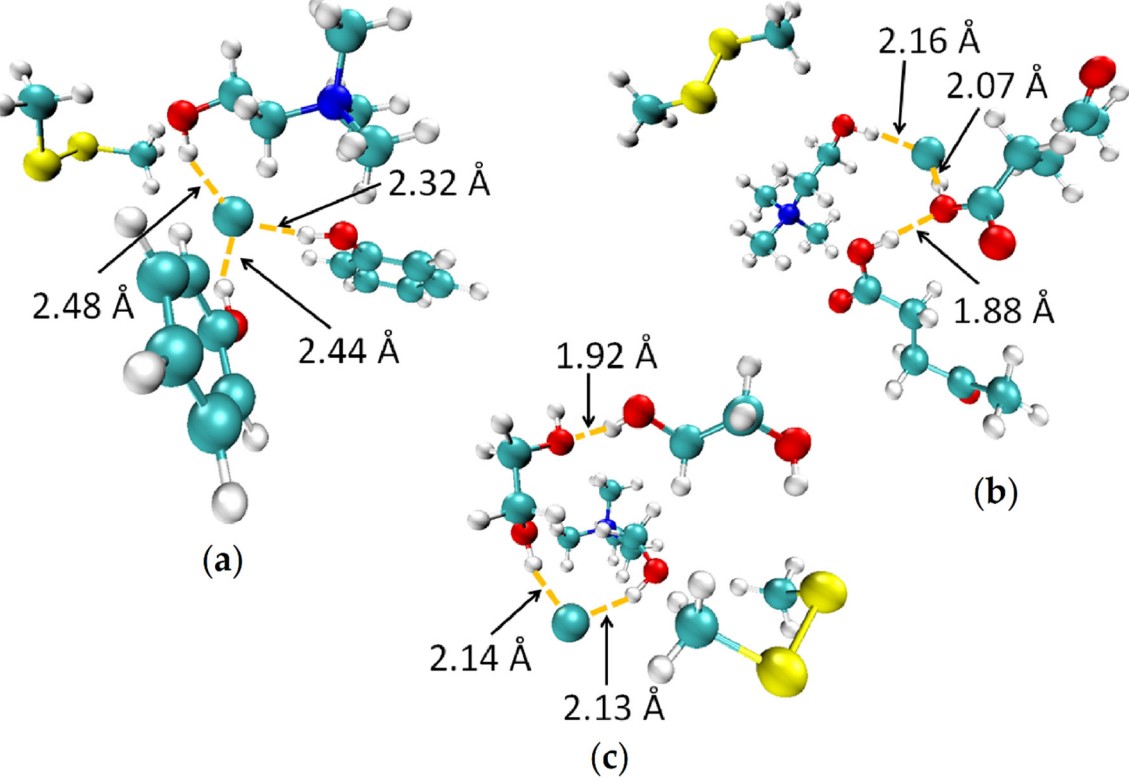

**Figure 4.** Optimized configurations of (**a**) ChCl:Ph-DMDS; (**b**) ChCl:Lev-DMDS; (**c**) ChCl:EG-DMDS. Carbon and chlorine atoms are displayed in light blue, nitrogen atoms are in dark blue, oxygen atoms are in red, sulfur atoms are in yellow, and hydrogen atoms are in white.

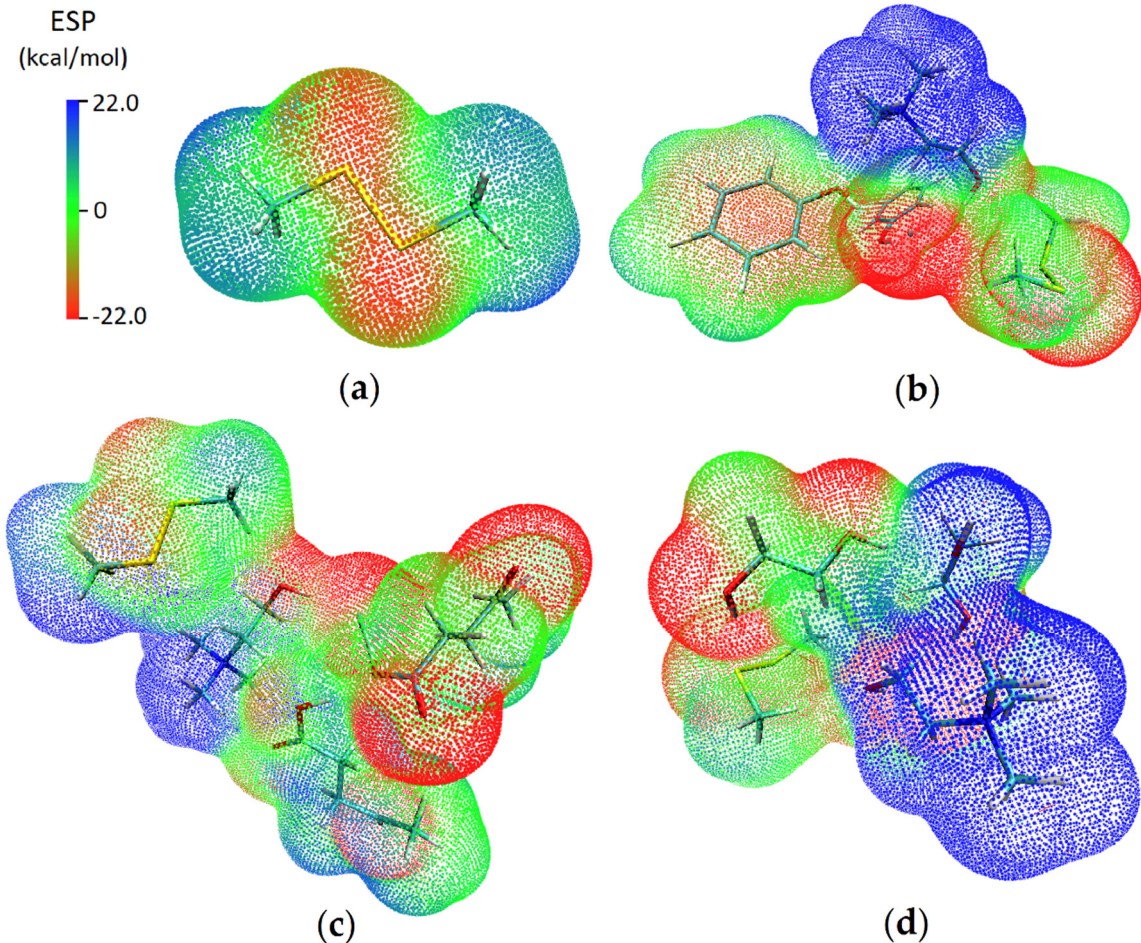

**Figure 5.** Electrostatic potential mapped on electron total density with an isovalue 0.001 for (**a**) DMDS; (**b**) ChCl:Ph-DMDS; (**c**) ChCl:Lev-DMDS; (**d**) ChCl:EG-DMDS. Blue color represents positive charges, red color represents negative charges.

The reduced density gradient (RDG) is a beneficial approach to distinguish and visualize various types of noncovalent interactions in real space. In the studies, the RDG analysis was used to visualized weak noncovalent interactions (i.e., hydrogen bond, van der Waals interaction, and repulsive effect), by plotting the RDG versus the electron density multiplied by the sign of the second Hessian eigenvalue, based on previous studies [44]. In Figure 6b,d,f, the green surfaces indicate van der Waals interaction, red surfaces indicate strong repulsion, and blue surfaces indicate an H-bond. The obtained data show that the three hydrogen bonds, as well as the van der Waals interaction, were formed between HBA and HBDs in all studied DES, which correspond to a large, negative $sign(\lambda2)\rho$ value (from $-0.04$ to $-0.02$ au) and $0.01$ au $< sign(\lambda2)\rho < 0.01$ au, respectively, in 2D diagrams (Figure 6a,c,e). Furthermore, in ChCl:Ph-DMDS, strong repulsive bonds occur ($sign(\lambda2)\rho = 0.02$ au), due to the presence of an aromatic ring in the phenol molecule. In all the studied complexes, between DES and DMDS, only a van der Waals interaction occur. The surfaces of the van der Waals interaction increase following the order of ChCl:EG-DMDS < ChCl:Lev-DMDS < ChCl:Ph-DMDS. This indicates that the main driving force affecting DMDS removal from model biogas is the van der Waals interactions.

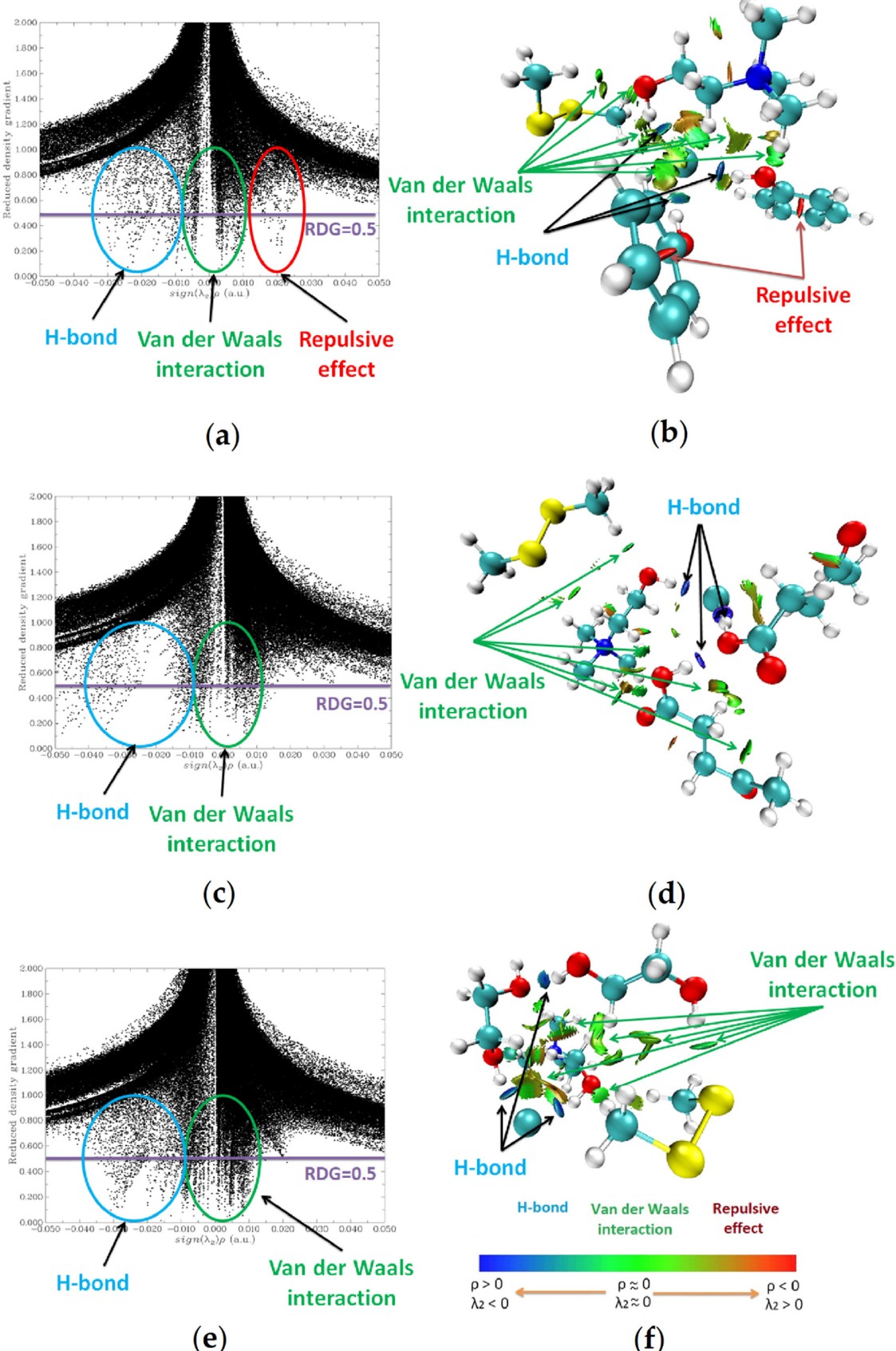

**Figure 6.** RDG isosurfaces (s = 0.5 a.u.) and 2D diagrams of electron density and its reduced density gradient for (**a**,**b**) ChCl:Ph-DMDS; (**c**,**d**) ChCl:Lev-DMDS; (**e**,**f**) ChCl:EG-DMDS.

The calculated interaction energy in the gas phase between DES and DMDS were −10.8, −3.7, and −9.6 kcal/mol for ChCl:Ph-DMDS, ChCl:Lev-DMDS, and ChCl:EG-DMDS, respectively. The lower interaction energy values stand for stronger interaction between DES and DMDS. The obtained data followed a similar trend to the experimental data: ChCl:Ph-DMDS < ChCl:EG-DMDS < ChCl:Lev-DMDS.

### 3.3. Regeneration and Reusability of DES

From an industrial point of view, the regeneration of DES is an essential and significant factor because it has a great impact on the operating cost. Therefore, regeneration of DES was carried on through one of the best-known regenerative methods, i.e., nitrogen barbotage (NG, which was carried for 2.5 and 5 h) and the adsorption process (with different types and amounts of adsorbents). In the adsorption process, three types of adsorbents were tested including SG, AC, and AO, in the amounts of 160 and 420 mg, which were added to 4 mL of each DES and shaken for 30 min. The results indicate that the nitrogen barboage is the most efficient desorption method. Desorption efficiency of DMDS from all DES after 5 h is higher than 99.999%. From ChCl:Ph, DMDS can be completely removed after 3 h (Figure 7).

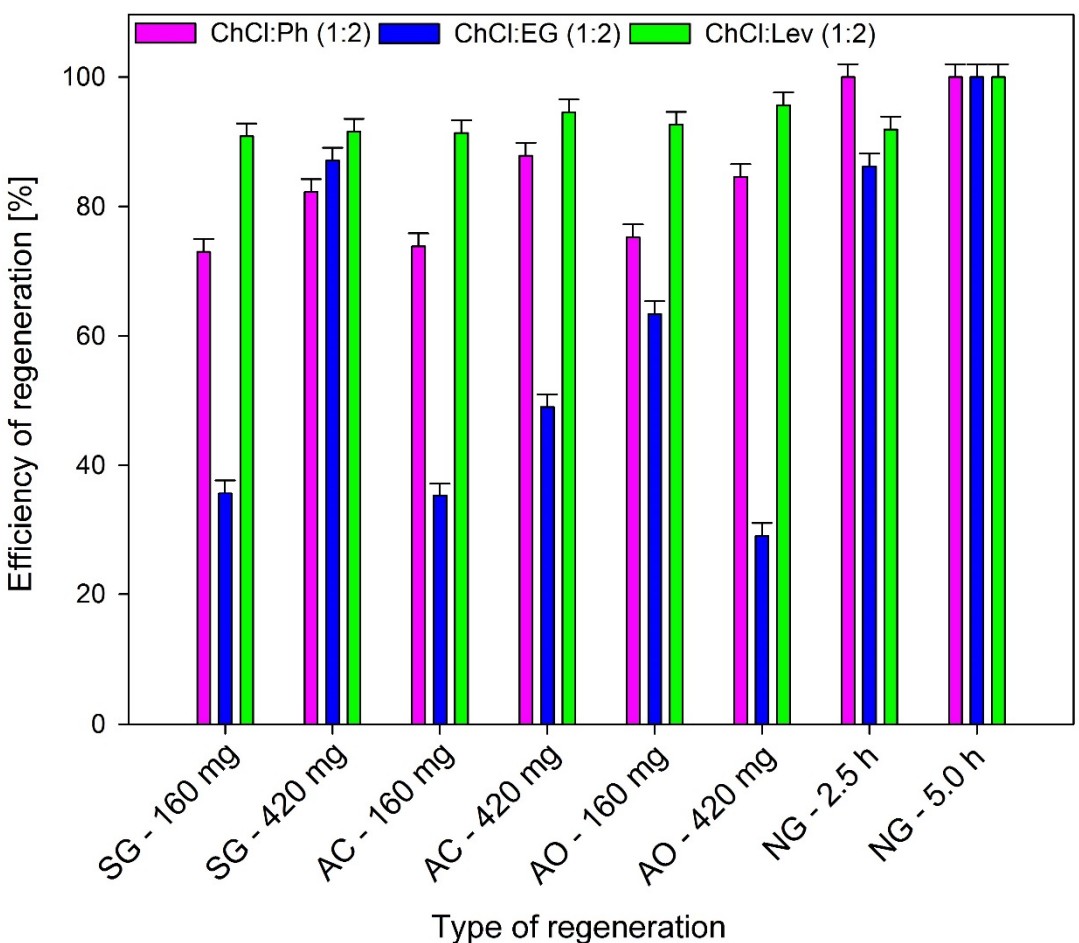

**Figure 7.** Regeneration efficiency of DESs after different regeneration methods.

For the remaining DES, this time should be extended, which would affect the cost of the process. In addition, the results show that slightly lower regeneration efficiencies for all DES were obtained using the adsorption process. The regeneration efficiency were in the range of 87.8%–84.5% for ChCl:Ph, 87.1%–63.4% for ChCl:EG, and 96.6%–91.3% ChCl:Lev. The highest absorption efficiency was obtained for 420 mg of SG. In most cases, the increase in adsorbent mass relative to the amount of DES containing

DMDS resulted in increased adsorption efficiency due to the increased adsorption surface. However, from an economical industrial point of view, the adsorbent amount should be as low as possible. Desorption is the most effective method to remove DMDS from ChCl:Lev. This is probably due to the low sorption capacity of ChCl:Lev and the absorption relatively small amount of DMDS.

The absorption−desorption results indicate that DMDS could be completely removed from DES and the absorbent could be reused for a minimum of five times without significant loss of absorption capacity DMDS (Figure 8a). In order to examine whether there were any structural changes in DES after the regeneration process (nitrogen barbotage), FT-IR analysis was used (Figure 8b–d). No additional peaks or shifts are observed in the DES spectra before and after regeneration, which indicates DES stability during the regeneration process.

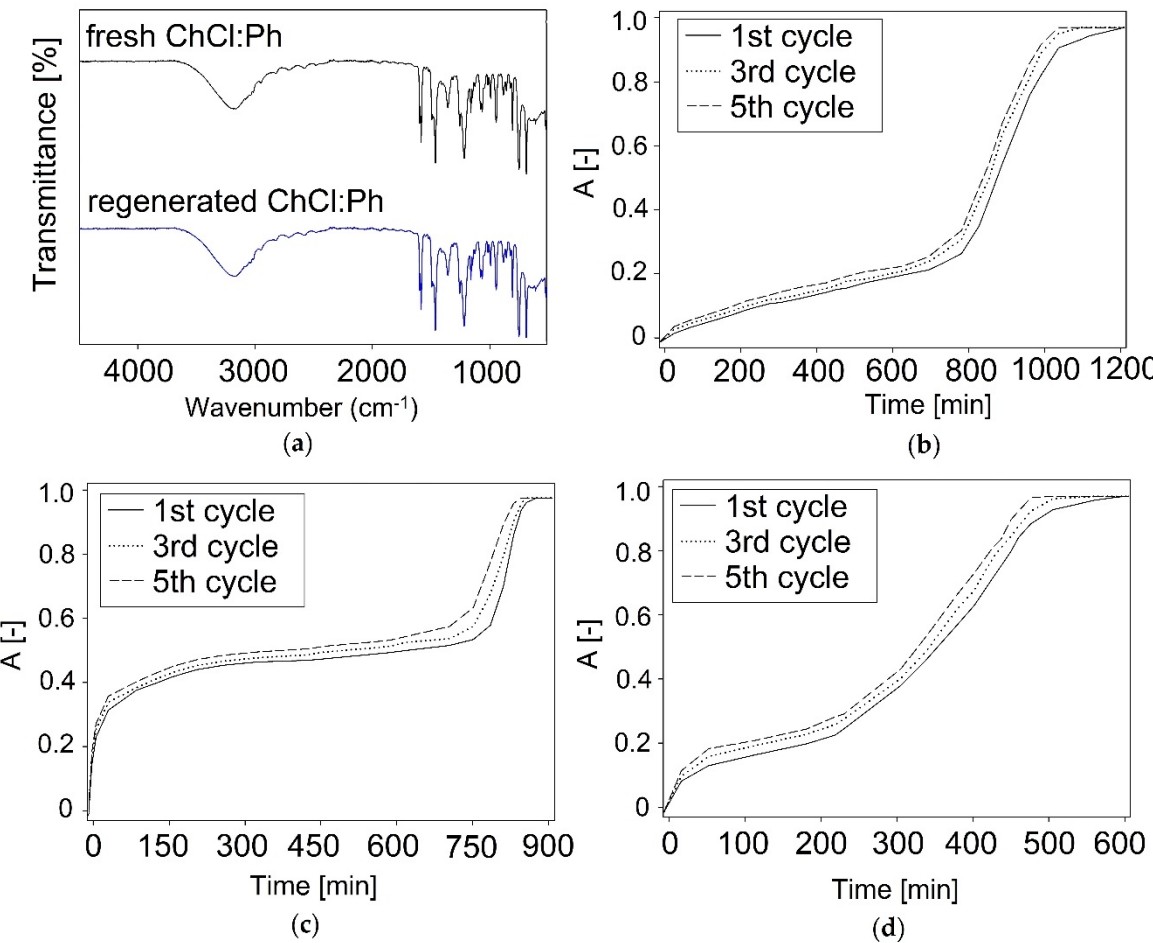

**Figure 8.** (**a**) FT-IR spectra recorded from on fresh ChCl:Ph, regenerated ChCl:Ph, and the reusability of DES: (**b**) ChCl:Ph; (**c**) ChCl:Lev; (**d**) ChCl:EG.

## 4. Conclusions

The choline chloride-based deep eutectic solvents have been synthesized and tested as alternative eco-friendly and green solvents for absorptive desulfurization of model biogas. Effect of selected absorption parameters including kind of DES, temperature, absorbent volume, model biogas flow rate, and initial concentration of DMDS were studied. It was found that the optimum absorption parameters for DMDS removal were absorption solvent ChCl:Ph in 1:2 molar ratio, 50 mL of DES, temperature 25 °C, and 50 mL/min flow rate. The influence of DMDS concentration indicates that the initial amount of DMDS in model biogas has only a minor effect on the absorption capacity and rate. In the optimum conditions, DMDS was removed with high efficiency for 800 min. After this time, the

gradual saturation of DES occurred. After the absorption process, ChCl:Ph in 1:2 molar ratio could be regenerated by means of nitrogen barbotage and reuse without loss absorption capacity.

The studies on the absorptive desulfurization mechanism indicate that the van der Waals interaction is the main driving force for the efficient removal of DMDS from model biogas.

The developed absorption process with the choline chloride-based deep eutectic solvents provides a promising alternative method for the removal of volatile organosulfur compounds from the real biogas stream.

The paper presents preliminary results of research on the removal of DMDS from a model biogas stream. However, due to the rich composition of real biogas samples, other volatile organic compound groups should also be included in future studies. In addition, in the future, to verify the suitability of the developed method, the studies using real samples of heavily contaminated biogas streams from sewage treatment plants and landfills will be carried out.

**Author Contributions:** Writing—original draft preparation, E.S. and P.M.; writing—review and editing, E.S. and P.M.; conceptualization and methodology, E.S. and P.M.; experimental set-up and software, E.S. and P.M.; conducted the experiments and data curation, E.S. and P.M. All authors have read and agreed to the published version of the manuscript.

**Funding:** This research received no external funding.

**Conflicts of Interest:** The authors declare no conflicts of interest. The funders had no role in the design of the study; in the collection, analyses, or interpretation of data; in the writing of the manuscript, or in the decision to publish the results.

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
