# Peer review of "Absorptive Desulfurization of Model Biogas Stream Using Choline Chloride-Based Deep Eutectic Solvents"

_sustainability, doi:10.3390/su12041619_

Round 1
Reviewer 1 Report
The authors described synthesis of deep eutectic solvents (DESs) and application them as absorption solvents for removal of dimethyl disulfide from (DMDS) from model biogas steam. The main parameters affecting the absorption capacity was carefully optimized, as well as reusability and regeneration of DESs was studied. Furthermore, mechanism of absorption was studied by means of density functional analysis (DFT). The presented paper is within the scope of the special issue of Sustainability and I think, the topic is interesting and contains many novelties. However the authors should make minor editorial corrections such as:
- In the conclusions, authors should add future research directions in the topic,
- Line 68: Authors should change the font size,
- Figure 3: the charts should be enlarged and would look better one after the others,
- Line 293: authors should expand the abbreviation CC,
- Line 330: authors should add full abbreviation of DES,
- Figure 5 is not mentioned anywhere in the main text.
Author Response
Dear Reviewer,
Thank you for your consideration and evaluation of the paper. All corrections raised by the Reviewer were thoroughly considered. Detailed explanations can be found below.
- In the conclusions, authors should add future research directions in the topic,
Authors add research directions in the topic according to the Reviewer's comment.
„The paper presents preliminary results of research on the removal of DMDS from a model biogas stream. However, due to the rich composition of real biogas samples, other volatile organic compound groups should also be included in future studies. In addition, in the future, to verify the suitability of the developed method, the studies using real samples of heavily contaminated biogas streams from sewage treatment plants and landfills will be carried out.”
- Line 68: Authors should change the font size,
The font size was corrected according to the Reviewer comment.
- Figure 3: the charts should be enlarged and would look better one after the others,
Figure 3 was appended according to the Reviewer's comment
- Line 293: authors should expand the abbreviation CC,
The manuscript was appended according to the Reviewer's comment
- Line 330: authors should add full abbreviation of DES,
The manuscript was appended according to the Reviewer's comment
- Figure 5 is not mentioned anywhere in the main text.
The manuscript was appended according to the Reviewer's comment
Thank you very much for your remarks in reviewing our work. We appreciate it very much, and we hope that we were able to satisfy the Reviewer.
Reviewer 2 Report
The manuscript includes a lot of work and contains many data that are originally synthesized and interpreted.
Comments:
- Studies on real samples might be of particular interest.
- There are many typos in the text of paper.
Author Response
Dear Reviewer
Thank you for your consideration and evaluation of the paper. All corrections raised by the Reviewer were thoroughly considered. Detailed explanations can be found below.
- Studies on real samples might be of particular interest.
We agree with Reviewer that the studies on a real samples could provide the interesting results. Therefore, we will consider the suggestion in the future studies. The detailed plans for the future are included in the conclusions: “The paper presents preliminary results of research on the removal of DMDS from a model biogas stream. However, due to the rich composition of real biogas samples, other volatile organic compound groups should also be included in future studies. In addition, in the future, to verify the suitability of the developed method, the studies using real samples of heavily contaminated biogas streams from sewage treatment plants and landfills will be carried out.”
- There are many typos in the text of paper.
The manuscript was trouble-checked for language typos, to make all statements clear for the readers.
Thank you very much for your remarks in reviewing our work. We appreciate it very much, and we hope that we were able to satisfy the Reviewer.